# Nuclear Membrane Protein SUN5 Is Highly Expressed and Promotes Proliferation and Migration in Colorectal Cancer by Regulating the ERK Pathway

**DOI:** 10.3390/cancers14215368

**Published:** 2022-10-31

**Authors:** Xiaoyue Song, Ruhong Li, Gang Liu, Lihua Huang, Peng Li, Wanjiang Feng, Qiujie Gao, Xiaowei Xing

**Affiliations:** 1Center for Experimental Medicine, Third Xiangya Hospital, Central South University, Changsha 410013, China; 2Department of Laboratory Medicine, Third Xiangya Hospital, Central South University, Changsha 410013, China; 3Department of General Surgery, Yanan Hospital Affiliated to Kunming Medical University, Kunming 650051, China; 4The Institute of Reproduction and Stem Cell Engineering, School of Basic Medical Sciences, Central South University, Changsha 410078, China

**Keywords:** SUN5, proliferation, migration, colorectal cancer, ERK1/2

## Abstract

**Simple Summary:**

Globally, colorectal cancer (CRC) is one of the most common malignant tumors, and the SUN protein family regulates the proliferation and migration of tumors. However, the biological function and mechanism of SUN5 in CRC are unclear. In this study, we found that SUN5 was highly expressed in CRC tissues and cells. Moreover, SUN5 promoted the proliferation and migration of CRC both in vitro and in vivo by cooperating with Nesprin2 to regulate the ERK pathway and interacting with Nup93 to promote the nuclear translocation of phosphorylated ERK1/2. Taken together, our findings provide novel insight into the role of SUN5 in CRC progression and imply that SUN5 might be a biomarker and therapeutic target for CRC patients.

**Abstract:**

SUN5 was first identified as a nuclear envelope protein involved in spermatocyte division. We found that SUN5 was highly expressed in some cancers, but its function and mechanism in cancer development remain unclear. In the present study, we demonstrated that SUN5 was highly expressed in colorectal cancer (CRC) tissues and cells, as indicated by bioinformatics analysis, and SUN5 promoted cell proliferation and migration in vitro. Moreover, the overexpression of SUN5 upregulated phosphorylated ERK1/2 (pERK1/2), whereas the knockdown of SUN5 yielded the opposite results. PD0325901 decreased the level of pERK1/2 to inhibit cell proliferation and migration, which was partially reversed by SUN5 overexpression, indicating that drug resistance existed in patients with high SUN5 expression. The xenograft transplantation experiment showed that SUN5 accelerated tumor formation in vivo. Furthermore, we found that SUN5 regulated the ERK pathway via Nesprin2 mediation and promoted the nuclear translocation of pERK1/2 by interacting with Nup93. Thus, these findings indicated that highly expressed SUN5 promoted CRC proliferation and migration by regulating the ERK pathway, which may contribute to the clinical diagnosis and new treatment strategies for CRC.

## 1. Introduction

Colorectal cancer (CRC) is one of the most common malignant tumors and the leading cause of cancer-related death [1]. According to global cancer statistics data in 2020 [2], around 1,100,000 new cases of CRC and 576,000 deaths from CRC are reported each year, especially since CRC is the third most common cancer in China [3]. Despite tremendous advances in chemotherapy and surgery on CRC, survival outcomes for CRC patients are far from ideal. Thus, it is essential to explore the pathogenesis of CRC to facilitate the development of potential diagnostic biomarkers and effective therapeutic targets.

The nuclear membrane is important in regulating cell migration, cell division, gene expression, and signaling pathways [4]. The LINC (linker of nucleoskeleton and cytoskeleton) complexes are formed by the SAD1/UNC84 domain (SUN) proteins interacting with the KASH (Klarsicht/ANC-1/Syne Homology) domain proteins and are involved in regulating the proliferation [5] and migration of tumors [6]. Five members have been identified in the SUN protein family, including SUN1, SUN2, SUN3, SUN4, and SUN5 (also termed SPAG4L) [7,8,9,10]. Matsumoto et al. demonstrated that SUN1 and SUN2 are downregulated in breast cancers [11]. In colorectal cancer, SUN2 inhibited cell migration and invasion by decreasing the expression of brain-derived neurotrophic factor (BDNF) to inhibit the BDNF/TrkB signaling pathway [12]. Different from SUN1 and SUN2, SUN4 was enriched in the testes and highly expressed in renal cell carcinoma [13], lung carcinoma [14], and hepatocellular carcinoma [15]. Knaup et al. demonstrated that SUN4 promoted migration and invasion by co-localizing with GM130 in renal clear cell carcinoma [13]. In a previous study, we found that SUN4 was highly expressed in lung cancer, and the knockdown of SUN4 could significantly inhibit the proliferation and migration of lung cancer cells [14]. Further bioinformatics analysis revealed that SUN1 and SUN2 probably originated from a close common ancestor, while SUN4 and SUN5 may stem from another origin [9]. The expression and function of SUN5 in tumors are currently unknown.

SUN5 gene was first cloned and submitted to GenBank by our team (GenBank accession AF401350), which was identified as a testis-specific expression gene [9,16]. The protein encoded by SUN5 was a nuclear membrane protein and had typical characteristics of the SUN protein family, including the transmembrane (TM) region at the N-terminus, coiled-coil (CC) domain, and conserved SUN domain at the C-terminus [9]. In the previous study, we established Sun5 knockout mice and found that deficiency of Sun5 disrupted the sperm head-to-tail connection due to the damage of the Sun5/Nesprin3 complex [17]. Moreover, in the meiosis stage, Sun5 was involved in spermatocyte division and nuclear migration [9]. Bioinformatics analysis revealed that SUN5 was expressed in various tumors and was higher in CRC tissues than in adjacent tissues. Thus, we speculated that SUN5 might play an important role in the progression of CRC.

The development of CRC was associated with abnormal activation of multiple signaling pathways [18,19]. The mitogen-activated protein kinases (MAPK) pathway, which consisted of a large family of serine-threonine kinases, was the major pathway related to cell proliferation and migration from the cell surface to the nucleus [20]. The extracellular-signal-regulated protein kinases (ERK) pathway was the most classical one in the MAPK pathway, which was involved in the pathogenesis, progression, and metastasis of CRC [21]. Activation of the ERK pathway was characterized by increased levels of phosphorylated ERK1/2. Warren et al. discovered that variant 2β of Nesprin2 (amino acids 1~125 (Spectrin Repeat 4, SR4) and 126~219 (SR3)) could interact directly with ERK1/2 [22]. Nesprin2 was the second member of KASH proteins, which could interact with SUN proteins to form LINC complexes [23]. Since bioinformatics analysis revealed that SUN5 could interact with Nesprin2, we speculated that SUN5 promoted the ERK pathway by cooperating with Nesprin2. Activated ERK1/2 were translocated into the nucleus and, in turn, performed their functions, including apoptosis, proliferation, invasion, and metastasis [21]. Phosphorylated ERK1/2 were usually transported by the nuclear pore complex (NPC) [24]. The NPC consisted of various nucleoporins [25], such as Nup93, Nup153 and ect. Nup93 was shown to transport phosphorylated ERK1/2 from the cytoplasm to the nucleus in breast cancer [26]. Thus, we hypothesized that in CRC, nuclear translocation of phosphorylated ERK1/2 might be mediated by Nup93.

In the present study, we investigated the expression of SUN5 in paired tissue samples from clinical CRC and used the SUN5 overexpression and knockdown cells to determine its function in vitro and in vivo. Finally, we explored the molecular mechanism of SUN5 to provide a new diagnostic marker and candidate therapeutic target for CRC.

## 2. Materials and Methods

### 2.1. Cell Lines and Cell Culture

Colorectal cancer cell lines were purchased from American Type Culture Collection (ATCC). SW620 and SW480 cells were cultured in L15 medium (KeyGEN BioTECH, Nanjing, China) and supplemented with 10% fetal bovine serum (FBS; Bio-Channel, Nanjing, China). LoVo cells were cultured in Dulbecco’s modified Eagle’s medium (DMEM; Gibco, Carlsbad, CA, USA) and supplemented with 10% FBS. DLD-1 cells were cultured in RPMI 1640 medium (Roswell Park Memorial Institute 1640; Gibco, Carlsbad, CA, USA) and supplemented with 10% FBS. All cells used in this study were maintained in a 5% CO_2_ cell culture incubator at 37 °C.

### 2.2. Collection of Tissue Samples

Paired samples of colorectal cancerous and paracarcinomatous tissues were obtained from 40 patients in the Department of Gastrointestinal Surgery, Third Xiangya Hospital of Central South University, Changsha, China, from December 2021 to June 2022. The ethics committee of Third Xiangya Hospital approved this study. None of the patients had undergone preoperative intervention therapy or chemotherapy. The tissue samples were immediately stored in liquid nitrogen for Western blotting and 10% formalin for immunohistochemistry (IHC). Detailed clinical data, including tumor size, clinical grade (according to tumor-node-metastasis (TNM) stage), histological type, differentiation degree, and lymph node metastasis were collected and statistically analyzed. 

### 2.3. Overexpression of SUN5

The full-length coding sequence of SUN5 with Flag tag was cloned into pLVX-IRES-Puro lentivirus vector to generate a pLVX-IRES-Puro-SUN5-Flag fusion expression plasmid. The recombinant and control plasmid were packaged into lentivirus, and the viral titers were greater than 10^8^ TU, respectively. Lentiviruses were co-cultured with cells for 12~18 h in the presence of polybrene. After 72 h of transfection, stably transfected cells were maintained in medium containing 2 μg/mL puromycin. SUN5 expression was determined for each group using Western blotting.

### 2.4. Knockdown of SUN5

A short hairpin RNA (shRNA) sequence against the SUN5 gene was designed and inserted into pLV3ltr-ZsGreen-Puro-U6 vector. The recombinant plasmid was packaged into lentivirus by Tsingke Biotech (Changsha, China). To establish stable SUN5 knockdown cell lines, LoVo and DLD-1 cells were infected with lentivirus containing the SUN5-shRNA sequence according to the manufacturer′s recommendations. After 72 h of transfection, cells were observed under a fluorescence microscope to evaluate infection efficiency and selected using 2 µg/mL of puromycin. The efficiency in different cells was determined by Western blotting. 

### 2.5. Silencing of Nesprin2

Nesprin2 silencing was completed by employing siRNAs specific for human Nesprin2 (called siRNA-Nesprin2) purchased from Tsingke Biotech. SUN5-overexpression LoVo cells were plated in 6-well plates and, at approximately 50% confluence, cells were transfected with siRNA-Nesprin2 and siRNA-control using lipofectamine 3000 (Thermo Fisher, Sunnyvale, CA, USA). Western blotting was performed at 72 h after transfection. The siRNA transfection experiments were repeated three times independently.

### 2.6. Western Blotting

Tissues were lysed in radioimmunoprecipitation assay (RIPA) buffer (Beyotime, Shanghai, China) with phenylmethanesulfonyl fluoride (PMSF) (Biosharp, Anhui, China) on ice after grinding with liquid nitrogen. Cells were collected and washed three times with phosphate-buffered saline (PBS) (KeyGEN BioTECH, Nanjing, China) and then lysed in RIPA with cocktail (Cellpro, Suzhou, China) on ice. After electrophoresis, the proteins were transferred to polyvinylidene fluoride (PVDF) membranes (Millipore, CA, USA). After blocking with 5% non-fat milk at room temperature for 2 h, the PVDF membranes were incubated with primary antibodies, including SUN5 (Thermo Fisher Scientific, Sunnyvale, CA, USA, 1:1000), β-tubulin (Affinity, Liyang, China, 1:10,000), pERK1/2 (Cell Signaling Technology, Danvers, MA, USA, 1:1000), and Nesprin2 (GeneTex, Irvine, CA, USA, 1:1000) at 4 °C overnight. After washing with TBST three times, horseradish-peroxidase-coupled mouse or rabbit secondary antibodies (Beyotime, Shanghai, China, 1:1000) were used for hybridization at room temperature for 1 h. Signals were visualized using an enhanced chemiluminescent (ECL) kit (Biosharp, Anhui, China), photographed, and measured using the VisionWorks system (Analytik Jena AG, Jena, Germany). The results are representative of at least three independent experiments.

### 2.7. Immunohistochemistry (IHC)

Human and mice tissues were fixed in 4% paraformaldehyde, embedded in paraffin, and sectioned. The immunostaining was performed according to the methods described by Du et al. [27]. Primary antibodies against SUN5 (Proteintech, Wuhan, China, 1:100) and pERK1/2 (Cell Signaling Technology, Danvers, MA, USA, 1:100) were used to detect their corresponding antigens. Images were captured using a light microscope (Zeiss, Oberkohen, Germany).

### 2.8. Cell-Proliferation Assays

The Cell Counting Kit-8 (CCK-8) assay (Cellpro, Suzhou, China) was used to measure cell proliferation in 96-well plates. Approximately 4000 LoVo and DLD-1 cells were seeded per well, with five replicates for each group. CCK8 reagents were added at 0, 24, 48, and 72 h, and incubated at 37 °C for 2 h. The absorbance values (OD450) were detected using an EnVision microplate reader (PerkinElmer, Waltham, MA, USA). We performed these experiments at least three times.

### 2.9. Colony-Formation Assays

For the colony-formation assays, approximately 300~500 LoVo or DLD-1 cells were seeded in each well of a 6-well plate in triplicate for each group and incubated for about two weeks. The colonies were fixed with methanol, stained with crystal violet, and counted by VisionWorks software (Analytik Jena AG, Jena, Germany). The average colony counts were calculated. The experiment was repeated in at least three replicates, and a paired *t*-test was used to test statistical significance. 

### 2.10. Wound-Healing Assay

Cells per well were grown to 80~90% confluence in 6-well plates. The cells were neatly scratched using sterile 200 μL pipette tips to form a wound area and maintained in serum-free medium. At least three photographs of randomly selected wound areas were taken within specific time points (0, 72 h) under a light microscope (Zeiss, Jena, Germany). The wound areas were analyzed by ImageJ software (Bethesda, MD, USA). All procedures were performed three times.

### 2.11. Transwell Assay

A Transwell assay was performed with filters with 8 μm-diameter pores (LabSelect, Beijing, China), with serum-free medium for the upper chamber and 30% FBS for the bottom chamber. Homogeneous single-cell suspensions were added to the upper chambers. After 48 h, the migrated cells on the bottom of the chambers were stained with crystal violet and counted in four random fields. The results were obtained from at least three independent experiments.

### 2.12. Identification of Differentially Expressed Genes (DEGs) Using RNA-Seq

The control and SUN5-overexpression cells were digested with trypsin, terminated by centrifugation, resuspended in TRlzol Reagent (Invitrogen, Shanghai, China), and freshly frozen in liquid nitrogen. The samples were sent to BGI-Wuhan (Wuhan, Hubei, China) Company for RNA extraction and eukaryotic transcriptome sequencing. The libraries were sequenced on an Illumina HiSeq (accessed on 26 February 2022, https://biosys.bgi.com/).

### 2.13. Real-Time Quantitative PCR Analyses

The total RNA from cells was extracted using RNAex Pro Reagent (Accurate Biology, Hunan, China). The first strand of cDNA was synthesized by reverse transcription of mRNA with the ReverTra Ace qPCR RT Master Mix (TOYOBO, Osaka, Japan). Quantitative real-time PCR (RT-qPCR) was performed using a LightCycler 480 Real Time PCR instrument (Roche, Basel, Switzerland). Actin was employed to normalize the RT-qPCR data. All the primer pairs were purchased from Tsingke Biotech, and the sequences are available in Appendix A. Each experiment was performed in three replicate wells and was repeated at least three times.

### 2.14. Nuclear and Cytoplasmic Fractionation

The cells were collected and precipitated using centrifugation (500× *g* for 10 min). The cells were washed with PBS three times and centrifuged at 1000 rpm for 5 min. According to the volume of the corresponding compacted cells, the corresponding volume of lysis buffer was added. The remaining steps were conducted according to the nuclear and cytoplasmic protein extraction kit protocol of Sangon Biotech (Shanghai, China). The experiment was repeated three times independently.

### 2.15. Cell Immunofluorescence Assay

Cell climbing sheets were prepared and fixed in 4% paraformaldehyde for 10 min at room temperature. Subsequently, permeabilization was performed with 0.5% Triton X-100 for 10 min at room temperature. Additionally, three washes were performed with PBS for 5 min each, and the solution blocked with 5% BSA for 2 h. Primary antibodies against Flag (Beyotime, Shanghai, China, 1:200), Nesprin2 (Abcam, Cambridge, UK, 1:200), ERK1/2 (Beyotime, 1:200), and pERK1/2 (Cell Signaling Technology, Danvers, MA, USA, 1:100) were incubated at 4 ℃ overnight. After washing with PBS three times, a fluorescent secondary antibody was used and incubated at room temperature for 1 h. The nuclei were stained with DAPI, blocked with glycerol, and finally observed under fluorescence microscope (Zeiss, Jena, Germany). Each process was repeated three times.

### 2.16. Co-Immunoprecipitation (Co-IP)

The total protein of the SUN5 stable overexpression cell line was extracted and lysed with IP lysis solution (Beyotime, Shanghai, China) with cocktail of phosphatase inhibitors (Cellpro, Suzhou, China) on ice. Antibodies were pre-bonded to magnetic beads at 4 ℃ for 6~8 h. After being properly lysed for 15 min, the complex was centrifuged for 30 min to separate the supernatant. Then, the total protein was incubated with the magnetic bead/antibodies complexes at 4 ℃ with rotation overnight, performed according to the instructions of Thermo Co-IP kit, and the Western blotting experiment was conducted on the purified protein subsequently obtained. The experiment was conducted in triplicate.

### 2.17. Inhibition of pERK1/2 Level with PD0325901

The level of phosphorylated ERK1/2 can be effectively inhibited by PD0325901 (MCE, Monmouth Junction, NJ, USA), which is a selective, non-ATP-competitive MEK inhibitor. Different drug concentrations were applied to obtain proper intervention conditions and the control group treated with DMSO. SUN5 overexpression and control LoVo or DLD-1 cells were cultured in medium supplemented with 1 µM or 10 µM of PD0325901, respectively, for 48 h, followed by experiments including CCK8, colony-formation, wound-healing, and Transwell assays. The results are representative of at least three independent experiments.

### 2.18. Tumor Xenograft

Female BALB/c nude mice (4–5 weeks, 18–20 g) were obtained from the Department of Laboratory Animals of Central South University and maintained under specific pathogen-free (SPF) conditions. All operations in this study were performed in accordance with the Guide for the Management of Laboratory Animals and Guide for the Welfare and Use of Animals in Cancer Research. This study was approved by the Ethics Committee of the Department of Laboratory Animals, Central South University (CSU-2022-0304). We first weighed the nude mice and randomly grouped them by ear tags. Then, 5 × 10^6^ LoVo SUN5 overexpression and control cells were subcutaneously injected into the left underarm of the nude mice (*n* = 4 per group). The length and width of the tumors were recorded daily using a caliper. Two weeks after the injection, the mice were sacrificed by an overdose of pentobarbital (250 mg/kg; intraperitoneal injection), and the tumor weight data were recorded. The tumor volumes were calculated based on the formula: volume (mm^3^) = length (mm) × width (mm) × width (as height) (mm)/2. The tumor tissues were further used for Western blotting and IHC.

## 3. Results

### 3.1. SUN5 Is Highly Expressed in Colorectal Cancer Tissues and Cells

Nuclear membrane proteins, especially SUN domain proteins, regulate the proliferation and migration of tumors [4]. The SUN5 gene was first cloned and identified by our team, which encodes a novel nuclear membrane protein, the fifth member of the SUN protein family [9]. The expression and function of SUN5 in tumors are currently unknown.

To investigate the expression of SUN5 in different tumor tissues, we collected clinical colorectal, liver, lung, esophageal, and gastric cancer tissues and found that SUN5 was detected in all cancer tissues, and especially highly expressed in CRC tissues (Figure 1A). Then, we analyzed SUN5 expression in 203 CRC tissues and 160 adjacent tissues from the GSE87211 cohort, and the result showed that the expression of SUN5 was significantly higher in CRC tissues than in adjacent tissues (*p* < 0.0001, Figure 1B). To further confirm this result, SUN5 expression was examined in 40 matched pairs of human CRC tissues and adjacent tissues by Western blotting. Statistical analysis of the blot results revealed that the expression of SUN5 was higher in CRC tissues than in adjacent tissues (*p* < 0.01, Figure 1C; partial results are shown in Figure 1D). Immunostaining results from two cases demonstrated that positive signals of SUN5 in CRC tissue cells were more obvious than in adjacent tissue cells (Figure 1E). Furthermore, we summarized and analyzed the clinicopathological characteristics of these 40 cases and found that the high SUN5 expression was associated with poor differentiation and lymph node metastasis (Appendix A). Immunofluorescence staining results showed that SUN5 was mainly distributed in the nucleus membrane and perinuclear cytoplasm in CRC cells (Figure 1F). Additionally, SUN5 was also detected in seven human CRC cell lines, including SW480, HT-29, LoVo, RKO, SW620, HCT116, and DLD-1, by Western blotting (Figure 1G). LoVo and DLD-1 cells were selected for subsequent functional assays.

### 3.2. SUN5 Promotes the Proliferation of Colorectal Cancer Cells 

To clarify the function of SUN5 in CRC, we established stable SUN5 overexpression cells (LoVo and DLD-1 transfected with pLVX-IRES-Puro-SUN5-Flag lentivirus) and knockdown cells (LoVo and DLD-1 transfected with PLV3ltr-ZsGreen-Puro-U6-shSUN5 lentivirus). Western blotting results showed that SUN5 was significantly upregulated in the SUN5-overexpression (SUN5-OE) group and downregulated in the SUN5-knockdown (SUN5-KD, also called shSUN5) group (*p* < 0.05, Figure 2A). Subsequently, the colony-formation assay was performed, and the results revealed that the number of colonies formed after 14 days was significantly increased in the SUN5-OE group and decreased in the SUN5-KD group compared with their corresponding control groups (Figure 2B). Meanwhile, we evaluated cell viability using a CCK-8 assay. The cell viability was higher in the SUN5-OE group and lower in the SUN5-KD group in LoVo cells at 72 h and DLD-1 cells at 48 h (*p* < 0.01, Figure 2C). Finally, we determined cell-cycle proteins related to proliferation by Western blotting. Compared with the control group, Cyclin D1, Cyclin B1, and CDK2 were upregulated in the SUN5-OE group and downregulated in the SUN5-KD group (*p* < 0.05, Figure 2D). These findings suggested that SUN5 promoted the proliferation of CRC cells.

### 3.3. SUN5 Promotes the Migration of Colorectal Cancer Cells 

To observe the effect of SUN5 on migration, a wound-healing assay was performed, and the results showed that the closure speed of the scratched area was significantly faster in the SUN5-OE group and slower in the SUN5-KD group at 72 h (Figure 3A,B). The Transwell assay, which evaluates cellular migration in three dimensions, further confirmed that the number of migrated cells significantly increased in the SUN5-OE group and decreased in the SUN5-KD group (Figure 3C). Epithelial–mesenchymal transition (EMT) enforced the ability of cells to metastasize and invade and could be measured by biomarkers [28], including E-cadherin, N-cadherin, and vimentin. In this study, N-cadherin and vimentin were upregulated in the SUN5-OE group and downregulated in the SUN5-KD group, while E-cadherin was downregulated in the SUN5-OE group and upregulated in the SUN5-KD group by Western blotting (*p* < 0.05, Figure 3D). These data indicated that SUN5 promoted the migration of CRC cells.

### 3.4. SUN5 Regulates the ERK Pathway in Colorectal Cancer

As seen in the data above, SUN5 can promote cell proliferation and migration in CRC. To further explore the molecular mechanisms, RNA-seq of SUN5-OE cells was performed, and the results demonstrated that 2613 genes were significantly upregulated (|log2FC| ≥ 1), while 115 genes were downregulated (|log2FC| ≥ 1) (Figure 4A,B). Gene ontology (GO) enrichment analysis indicated that SUN5 was involved in regulating biological processes, cellular components, and molecular functions (Figure 4C). Further, the Kyoto Encyclopedia of Genes and Genomes (KEGG) analysis showed that many of the proteins encoded by DEGs participated in multiple signaling pathways, such as MAPK, cell cycle, mTOR, Wnt, ubiquitin-mediated proteolysis, and Hippo signaling, which exert roles in proliferation, migration, and survival (Figure 4D). We identified six DEGs associated with cell proliferation and migration by RT-qPCR and found that CCNA2, CD164 [29], CKAP2 [30], and RAD21 [31] were upregulated in SUN5-OE cells, while CDKN1A and EGR1 were downregulated (*p* < 0.01, Figure 4E), which were consistent with RNA-seq results.

Based on RNA-seq and KEGG pathway enrichment analysis, we focused on the MAPK pathway. Since the ERK pathway was the classical signaling pathway in the MAPK pathway, we detected phosphorylated ERK1/2 and total ERK1/2 in the SUN5-OE, SUN5-KD cells, and their corresponding control cells, and the results showed that the ERK1/2 phosphorylation level was upregulated in SUN5-OE cells and downregulated in SUN5-KD cells, but the total level of ERK1/2 did not change compared with their corresponding control cells (*p* < 0.05, Figure 4F). These data suggested that overexpression of SUN5 could upregulate the levels of phosphorylated ERK1/2 and promote the ERK pathway.

### 3.5. Overexpression of SUN5 Attenuates the Effect of PD0325901 on Colorectal Cancer Cells

PD0325901, a MEK inhibitor, was used to block ERK pathway activation [32]. LoVo and DLD-1 cells were cultured in the medium supplemented with different drug concentrations (0.1, 1, 10, 50, and 100 µM) for 48 h. The level of pERK1/2 was inhibited by PD0325901 from 0.1 to 100 µM in LoVo cells and from 1 to 100 µM in DLD-1 cells (Figure 5A,B). The SUN5-OE LoVo and DLD-1 cells were cultured in the medium supplemented with the above drug concentrations for 48 h as well. Western blotting results revealed that PD0325901 was also able to effectively inhibit the level of phosphorylated ERK1/2 in SUN5-OE cell lines, but the inhibitory effect of increasing PD0325901 drug concentrations on ERK1/2 phosphorylation levels did not significantly increase (Appendix A), suggesting that overexpression of SUN5 partially attenuated the inhibition of ERK1/2 phosphorylation levels by PD0325901. Therefore, to effectively inhibit the pERK1/2 level with less impact on cell viability, 1 µM and 10 µM PD0325901 were used on LoVo and DLD-1, respectively, for the following experiments. 

NC and SUN5-OE LoVo cells were cultured in a medium supplemented with 1 µM of PD0325901 for 48 h. In NC and SUN5-OE cells, PD0325901 significantly inhibited cell viability and migration, indicating that the blockade of the ERK signaling pathway may result in the inhibition of proliferation and migration in CRC cells. However, in the PD0325901 group, the viability and migration were higher in SUN5-OE than in NC (*p* < 0.05, Figure 5C,D), suggesting that overexpression of SUN5 might attenuate the inhibition of proliferation and migration induced by PD0325901 in CRC. Similar results were also obtained for DLD-1 cells.

### 3.6. SUN5 Accelerates Tumor Growth In Vivo

To further evaluate the function of SUN5 in vivo, we injected SUN5-OE and control LoVo cells into nude mice. After two weeks, all the mice developed tumors at the injection site (Figure 6A,B), but the average size (Figure 6C) and weight of the tumors generated by the SUN5-OE group were significantly bigger than in the control group (*p* < 0.05; Figure 6D,E). However, there was no difference in body weight between the two groups, and the stable and continuous increase indicated that the mice were in good physical condition (*p* > 0.05; Figure 6F). Further immunostaining results demonstrated that positive signals of SUN5 and pERK1/2 in the SUN5-OE group were more obvious than in the control group (Figure 6G). Two tumor tissue samples were randomly selected from each group to detect the changes in downstream factors by Western blotting. The results revealed that the expression of E-cadherin was downregulated, and N-cadherin, vimentin, Cyclin D1, and pERK1/2 were upregulated in SUN5-OE tumor tissues (*p* < 0.05, Figure 6H,I), which is consistent with in vitro results. The results confirmed the promotion function of SUN5 in CRC in vivo.

### 3.7. SUN5 Promotes the ERK Pathway via Nesprin2 Mediation

The mechanism of SUN5-regulated phosphorylated ERK1/2 was previously unknown. Previous studies revealed that LINC complexes regulate multiple signaling pathways, including the ERK pathway and others [33]. The LINC complexes were comprised of the SUN domain proteins interacting with the KASH domain proteins (Nesprin1~4) [34]. SUN5 belongs to the SUN protein family, and bioinformatics analysis indicated that SUN5 might interact with Nesprin2 (Figure 7A). Warren et al. found that the variant of Nesprin2 could interact directly with ERK1/2 [22]. Thus, we focused on the interaction chain of SUN5, Nesprin2, and ERK1/2 and speculated that SUN5 regulated the ERK pathway by interacting with Nesprin2. To verify this hypothesis, cellular immunofluorescence (IF) staining was performed, and the results showed that SUN5 (labeled with red fluorescence) and Nesprin2 (labeled with green fluorescence) were co-localized in the nuclear membrane and cytoplasm, while ERK1/2 and pERK1/2 (all labeled with green fluorescence) were localized in the nucleus and cytoplasm (Figure 7B). Further co-immunoprecipitation (Co-IP) assays confirmed the interaction between SUN5, Nesprin2, and ERK1/2. The SUN5-OE cell proteins were subjected to IP with an anti-Flag antibody, anti-Nesprin2 antibody, and control IgG, respectively. The results revealed that SUN5 could cooperate with Nesprin2 to form the LINC complex in CRC. The Co-IP assay was also performed using an anti-Nesprin2 antibody and anti-ERK1/2 antibody, and a complex containing ERK1/2 and Nesprin2 was clearly detected in CRC cells, suggesting that Nesprin2 interacted with ERK1/2 in CRC (Figure 7C). Taken together, these data indicated that there was an interaction chain of SUN5/Nesprin2/ERK1/2 in CRC.

To further investigate the importance of Nesprin2 in regulating the ERK pathway by SUN5, we conducted a Nesprin2 silencing experiment on SUN5-OE cells. The results showed that the knockdown of Nesprin2 could significantly downregulate the phosphorylated ERK1/2 protein level but did not affect the total ERK1/2 (*p* < 0.05, Figure 7D). Thus, we speculated that silencing of Nesprin2 reduced SUN5/Nesprin2 complex formation, leading to the downregulation of phosphorylated ERK1/2 levels. Together, these findings indicated that SUN5/Nesprin2 and the LINC complex were essential for regulating the ERK pathway.

### 3.8. SUN5 Promotes Phosphorylated ERK Nuclear Translocation by Interacting with Nup93 

When the ERK pathway was activated, phosphorylated ERK levels were elevated and subsequently translocated into the nucleus. Phosphorylated ERK1/2 mainly accumulates in the nucleus to perform functions, including regulating gene transcription and promoting cell proliferation and migration [21]. In this study, we detected the expression of pERK1/2 in the cytoplasm and nucleus of CRC cells and observed that overexpression of SUN5 increased the nuclear accumulation of phosphorylated ERK1/2, with Lamin B1 as the internal reference in the nucleus and GAPDH as the internal reference in the cytoplasm (*p* < 0.05, Figure 8A). Further cellular immunofluorescence (IF) staining provided evidence that phosphorylated ERK1/2 (labeled green fluorescence) was abundantly presented in the nucleus, which co-localized with DAPI (labeled blue fluorescence) (Figure 8B). 

Previous studies revealed that phosphorylated ERK1/2 was associated with nucleoporins. Nataraj et al. demonstrated that nucleoporin 93 (Nup93) transported phosphorylated ERK1/2 from the cytoplasm to the nucleus by interacting with Improtin7 [26]. In our previous study, we performed Co-IP/MS (co-immunoprecipitation coupled to mass spectrometry) of total mouse testicular proteins to identify SUN5 interacting proteins and found that Nup93 was identified as a candidate interaction partner of SUN5. Thus, we speculated that SUN5 promoted phosphorylated ERK1/2 to the nuclear pore complex (NPC) by cooperating with Nesprin2 and accelerated the nuclear translocation of phosphorylated ERK1/2 by interacting with Nup93. Subsequently, we performed a Co-IP assay using specific antibodies to confirm the interaction between SUN5 and Nup93 in CRC. In the immunoprecipitants of anti-SUN5 antibody or anti-Nup93 antibody, both Nup93 and SUN5 were detected in SUN5-OE cells (Figure 8C). We suggested that the SUN5/Nup93 complex was important in the nuclear translocation of phosphorylated ERK1/2.

## 4. Discussion

In recent years, the incidence and mortality rate of colorectal cancer has remained high, and more research on the pathogenesis of CRC is essential. In the present study, we first demonstrate that SUN5 is highly expressed in clinical human CRC tissues and cells. Overexpression of SUN5 promotes cell proliferation and migration in vitro and accelerates tumor formation in vivo. Mechanistically, SUN5 regulates the ERK pathway via Nesprin2 mediation, suggesting that the SUN5/Nesprin2/pERK axis is important for the development of CRC. Moreover, we demonstrate that SUN5 accelerates the nuclear translocation of pERK1/2 by interacting with Nup93, which provides a novel insight into pERK1/2 nuclear translocation.

SUN domain proteins, as inner nuclear membrane proteins, regulate proliferation and migration in tumors [5,35,36]. SUN1 and SUN2 are downregulated in tumors and exert tumor-suppressive effects [11]. SUN4 is upregulated in tumors, which promotes cell proliferation and migration and has been considered a tumor marker [37]. In this study, we demonstrate that SUN5, the fifth member of the SUN protein family, is highly expressed in CRC tissues and cells. Overexpression of SUN5 promotes cell proliferation and migration, and knockdown of SUN5 inhibits these functions in CRC, which is similar to the functions of SUN4 in tumors but different from SUN1 and SUN2. Further analysis of clinical tissue data revealed that high SUN5 expression was associated with poor differentiation and lymph node metastasis, indicating that SUN5 may be a potential biomarker for CRC clinic diagnosis. 

Both mitosis and meiosis are forms of cell proliferation. Mitosis is the major proliferation type in eukaryotic cells, while meiosis is a special type of cell division in germ cells [38]. In our previous study, SUN5 was involved in spermatocyte division and nuclear migration in the meiosis stage [9]. Nuclear migration is a precondition for cell migration. Since the nucleus is mechanically connected to the cytoskeleton and eventually to the extracellular matrix (ECM) by adhesion, the nucleus alters morphology together with the cell during 3D migration [28]. In this study, we found that SUN5 promotes proliferation and migration in CRC. Further research on the role of SUN5 in drug sensitivity and self-renewal is essential, which is important for the diagnosis and treatment of colorectal cancer.

The development of CRC is mediated by the imbalance of multiple signaling pathways, such as MAPK, PI3K, Wnt, NOTCH, and TGF-β pathways [19]. In the present study, KEGG pathway enrichment of DEGs reveals that SUN5 regulates the MAPK signaling pathway. The results of cellular experiments showed that overexpression of SUN5 upregulates the level of phosphorylated ERK1/2, and knockdown of SUN5 inhibits its expression, suggesting that SUN5 promotes the ERK pathway in CRC. Moreover, the inhibition of the ERK pathway is considered a strategy for tumor treatment. PD0325901 is a MEK inhibitor and can effectively inhibit the level of phosphorylated ERK1/2. Scientists from the Dana Farber Cancer Institute reported that PD0325901 effectively treated Ras-driven lung cancer combined with Pabocillin (Annual Meeting of the American Association for Cancer Research in 2017). Here, we demonstrate that the proliferation and migration of CRC cells are significantly inhibited by the PD0325901 treatment. Overexpression of SUN5 attenuates the inhibition of proliferation and migration induced by PD0325901 in CRC. Based on these results, we suggest that drug resistance may exist in patients with high SUN5 expression. 

SUN domain proteins can interact with KASH domain proteins to form LINC complexes. LINC complexes play an important role in regulating cell proliferation, migration, and invasion. Ji et al. revealed that SUN4 interacts with Nesprin3 to promote lung cell proliferation and migration [14]. In our study, we demonstrate an interaction chain of SUN5, Nesprin2, and ERK1/2 in CRC. Warren et al. reported that the variant of Nesprin2 could interact directly with ERK1/2 [22]. Cell experiments reveal that Nesprin2 silencing inhibits the level of phosphorylated ERK1/2 in SUN5-OE cells, suggesting that decreased formation of the SUN5/Nesprin2 complex causes downregulation of pERK1/2. These results indicate that the SUN5/Nesprin2/pERK axis is important in the progression of CRC. 

Phosphorylated ERK1/2 is mainly accumulated in the nucleus to perform its functions. Unlike the canonical nuclear localization signal (NLS) and Imp α/β nuclear-shuttling machinery [39], nuclear translocation of pERK1/2 begins with nuclear translocation signal (NTS) exposure and phosphorylation, and phosphorylated NTS promotes pERK1/2 to interact with importin7, which subsequently escorts pERK1/2 to the nucleus via the nuclear pore [40]. The nuclear pore complex (NPC) is the sole bidirectional gateway regulating RNAs, proteins, and other macromolecules during nucleocytoplasmic transport [41]. Nataraj et al. revealed that nucleoporin93 (Nup93) transported pERK1/2 from the cytoplasm to the nucleus by interacting with improtin7 [26]. In this study, we demonstrated that SUN5 accelerated the nuclear translocation of pERK1/2 by interacting with Nup93. Regarding the more detailed mechanisms involved, we speculate that SUN5 may promote pERK1/2 to NPC via Nesprin2 mediation and interact with Nup93 to promote its nuclear translocation, which provides a novel insight into the nuclear translocation of pERK1/2. 

## 5. Conclusions

In summary, we demonstrate that SUN5 is highly expressed in CRC and promotes proliferation and migration through the ERK pathway. Our findings suggest that SUN5 regulates the ERK pathway via Nesprin2 mediation and promotes the nuclear translocation of phosphorylated ERK1/2 by interacting with Nup93, providing novel insights into the pathogenesis of CRC. 

## Figures and Tables

**Figure 1 cancers-14-05368-f001:**
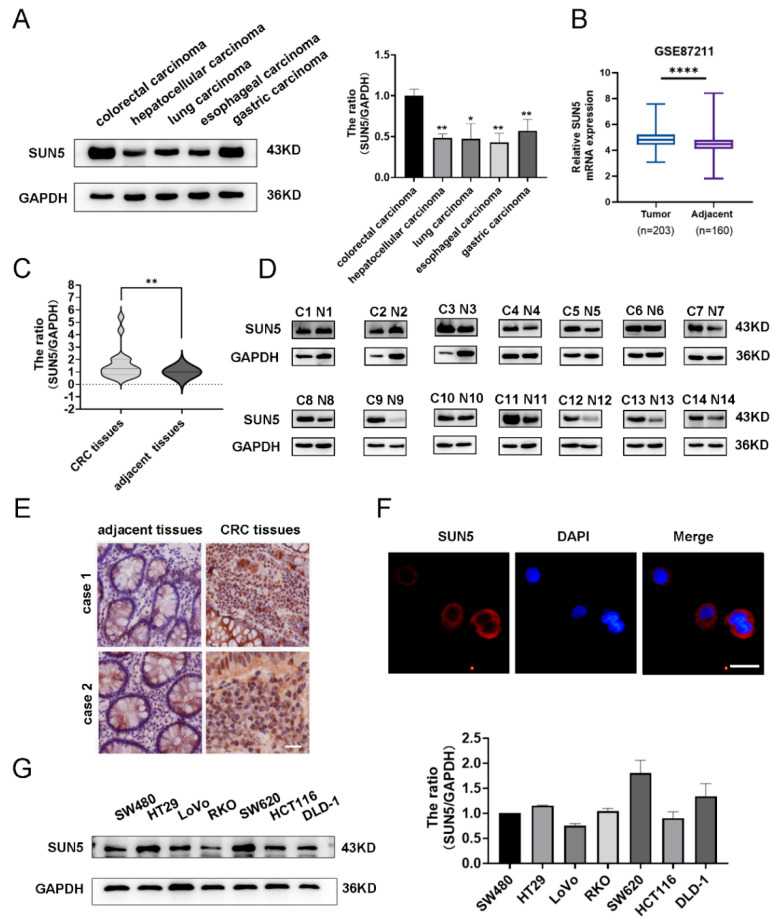
High expression of SUN5 was detected in colorectal cancer tissues and cells. (**A**) SUN5 protein levels in colorectal carcinoma, hepatocellular carcinoma, lung carcinoma, esophageal carcinoma, and gastric carcinoma (measured by Western blotting, * *p* < 0.05, ** *p* < 0.01). (**B**) The expression of SUN5 in 203 CRC tissues and 160 adjacent tissues according to GSE87211 cohort (**** *p* < 0.0001. Student’s *t*-test). (**C**) SUN5 protein levels in CRC tissues compared with those in adjacent tissues (*n* = 40, measured by Western blotting, ** *p* < 0.01). (**D**) Western blotting results of protein levels of SUN5 in colorectal cancer tissues and adjacent tissues in 14 cases; the “C” represents CRC tissues, and the “N” represents adjacent tissues. (**E**) Representative IHC images of SUN5 in human CRC tissues (right) and adjacent tissues (left). Scale bar: 30 μm. (**F**) Immunofluorescence staining results showed that SUN5 was mainly distributed in the nucleus membrane and perinuclear cytoplasm in DLD-1 cells. Scale bar: 20 μm. (**G**) SUN5 protein levels in CRC cell lines (SW480, HT29, LoVo, RKO, SW620, HCT116, and DLD-1) were analyzed by Western blotting.

**Figure 2 cancers-14-05368-f002:**
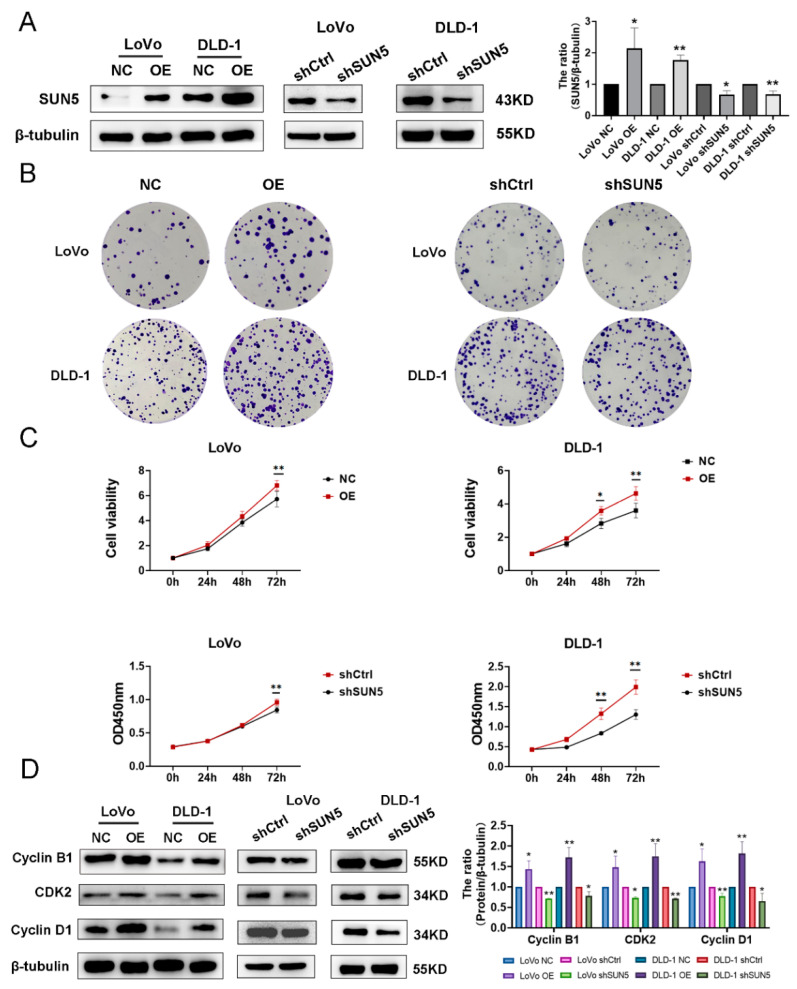
Overexpression of SUN5-promoted proliferation, while knockdown of SUN5 inhibited the proliferation of colorectal cancer cells in vitro. (**A**) Transfection of SUN5 overexpression (OE) lentivirus upregulated SUN5 expression when compared with the control lentivirus (NC), while shRNA silence of SUN5 (shSUN5, also called SUN5-KD) downregulated SUN5 expression when compared with the negative control shRNA (shCtrl) in LoVo and DLD-1 cells by Western blotting. β-tubulin was used as an internal control. (**B**,**C**) Overexpression or knockdown of SUN5 promoted or inhibited cell proliferation of CRC cells measured by colony-formation and CCK-8 assays (* *p* < 0.05; ** *p* < 0.01). (**D**) Cyclin D1, CDK2, and Cyclin B1 were determined by Western blotting after overexpression or knockdown of SUN5 in LoVo and DLD-1 cells (* *p* < 0.05, ** *p* < 0.01).

**Figure 3 cancers-14-05368-f003:**
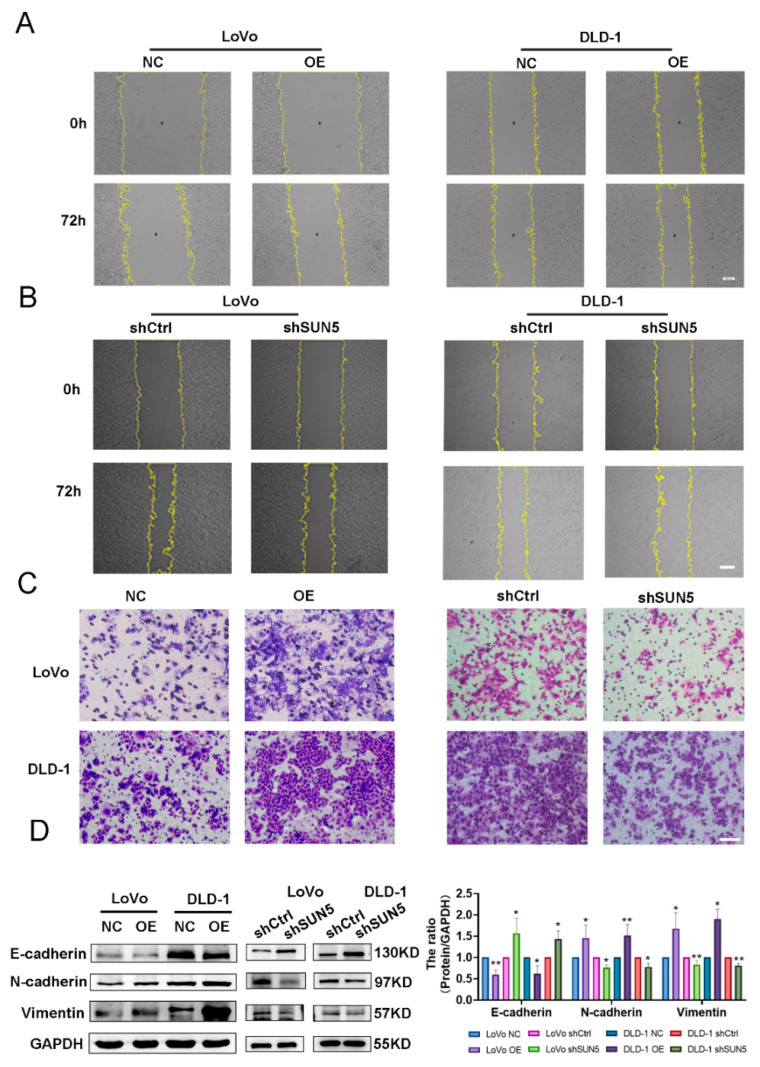
Overexpression of SUN5 promoted migration, while knockdown of SUN5 inhibited the migration of colorectal cancer cells in vitro. (**A**–**C**) Overexpression or knockdown of SUN5 promoted or inhibited cell migration as analyzed by wound-healing (scale bar: 200 μm) and Transwell assay (scale bar: 100 μm). (**D**) E-cadherin, N-cadherin, and vimentin were determined by Western blotting after overexpression or knockdown of SUN5 in LoVo and DLD-1 cells (* *p* < 0.05, ** *p* < 0.01).

**Figure 4 cancers-14-05368-f004:**
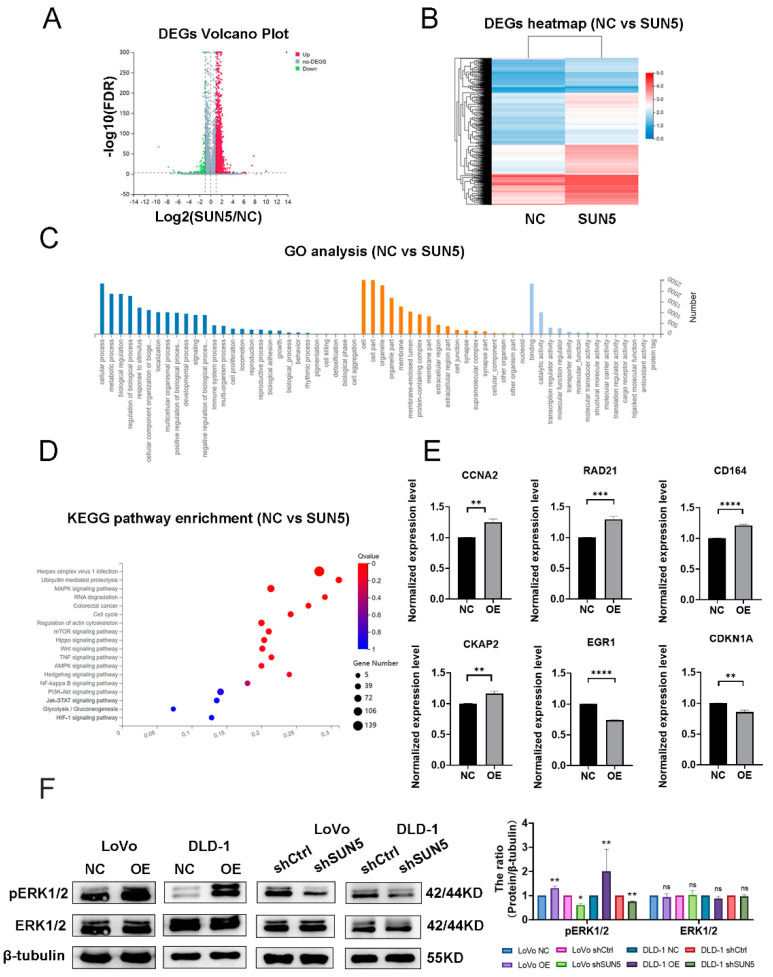
SUN5 regulated the level of phosphorylated ERK1/2 in colorectal cancer. (**A**) Scattered map of upregulated (red) and downregulated (green) genes positioned away from unchanged genes (gray). (**B**) Heatmap of DEGs comparing control and SUN5-OE group. (**C**) GO enrichment analysis of cellular components, molecular functions, and biological processes of DEGs comparing control and SUN5-OE group. (**D**) KEGG analysis showed that the proteins encoded by these DEGs were involved in multiple signaling pathways. (**E**) Six genes, including CCNA2, CD164, CDKN1A, CKAP2, EGR1 and RAD21, were identified by RT-qPCR to verify the creditability of the sequencing (Student′s *t*-test, *n* = 3. ** *p* < 0.01, *** *p* < 0.001, **** *p* < 0.0001). (**F**) ERK1/2 and the phosphorylation of ERK1/2 were determined by Western blotting after overexpression and knockdown of SUN5 in LoVo and DLD-1 cells (* *p* < 0.05, ** *p* < 0.01; “ns” indicates not significantly different).

**Figure 5 cancers-14-05368-f005:**
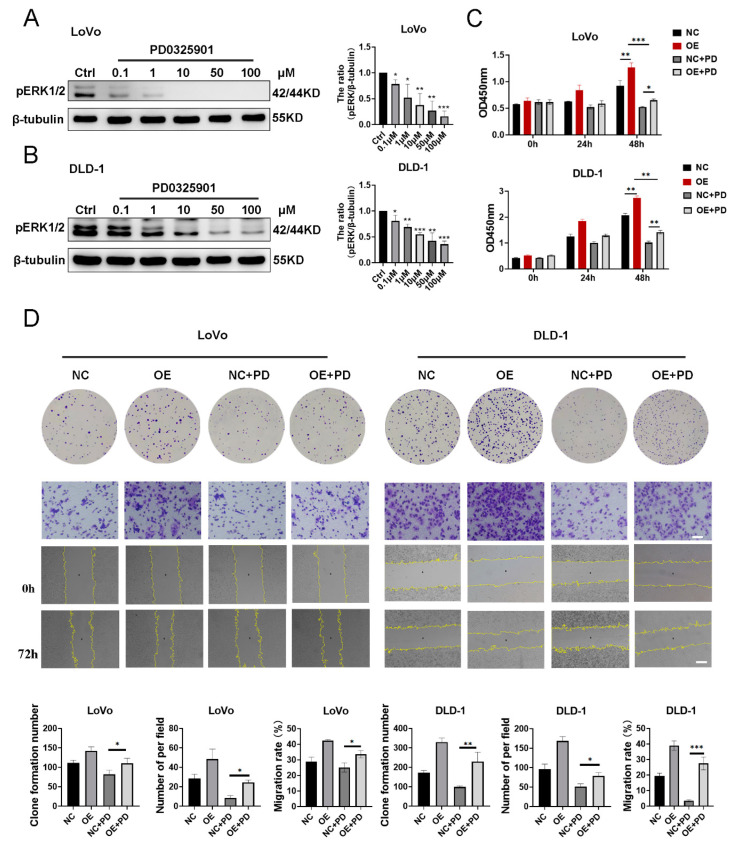
Overexpression of SUN5 partly attenuated the inhibition of proliferation and migration of CRC cells by PD0325901. (**A**,**B**) The Western blotting analysis of the pERK1/2 level in CRC cells treated with different concentrations of PD0325901 (0.1, 1, 10, 50, and 100 µM) for 48 h. “Ctrl” indicates the control group with the added solvent (DMSO). (**C**,**D**) The cell proliferation and migration ability were evaluated by CCK8 (two-way ANOVA); colony-formation (*t*-test) and Transwell (*t*-test); and wound-healing assays (two-way ANOVA), respectively (* *p* < 0.05, ** *p* < 0.01, *** *p* < 0.001). Scale bar: 100 μm (above), 200 μm (below).

**Figure 6 cancers-14-05368-f006:**
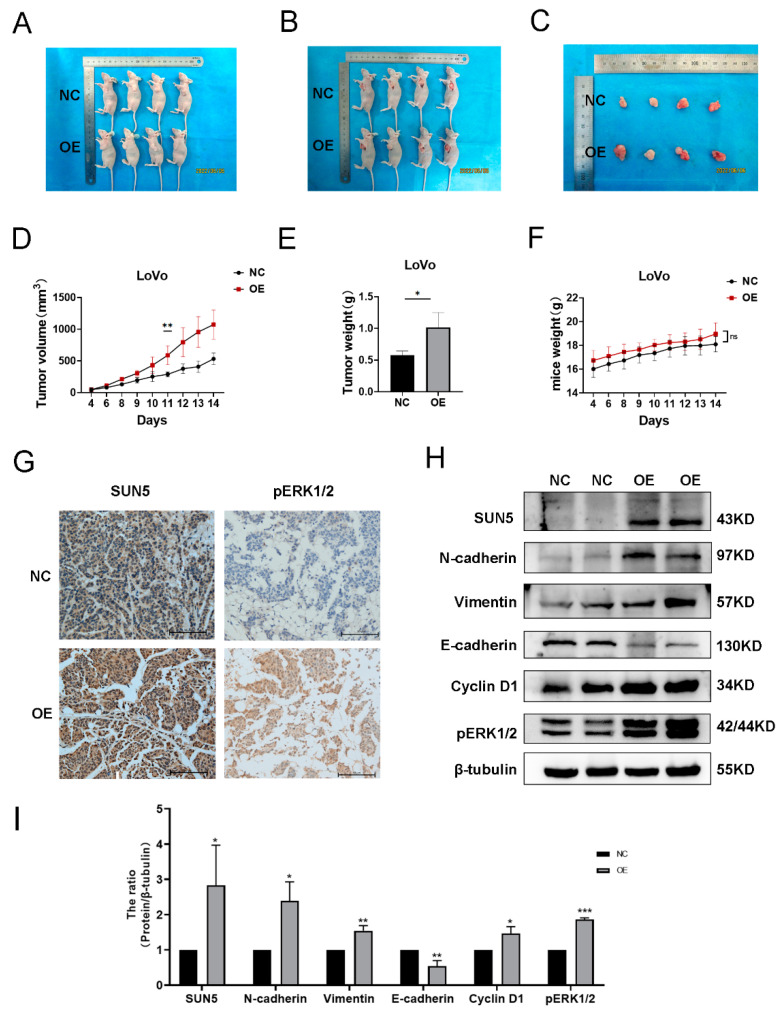
SUN5 overexpression accelerated the tumorigenicity of LoVo cells in vivo. (**A**) The subcutaneous tumor in nude mice formed by LoVo cells transfected with control and SUN5-OE lentivirus (*n* = 4 in each group). After two weeks, all the mice developed tumors at the injection site; whole-body imaging was performed for each nude mouse, and the representative pictures were recorded and presented. (**B**) Images of subcutaneous tumorigenic nude mice after skin cutting. (**C**) Images of CRC tumor xenografts from each mouse. (**D**) Tumor volumes were recorded and analyzed (** *p* < 0.01). (**E**) Tumor weights were evaluated (* *p* < 0.05). (**F**) Nude mice weights were recorded and analyzed. (**G**) IHC staining of tumor tissues verified that SUN5 and pERK1/2 signals were still significantly obvious in the SUN5-OE group at the examination time (two weeks after the xenograft). Scale bar: 50 μm. (**H**,**I**) Western-blotting assay showed the expression of SUN5, pERK1/2, Cyclin D1, E-cadherin, N-cadherin, and vimentin in tumor xenografts (in which two tumor tissue samples were randomly selected from each group; * *p* < 0.05, ** *p* < 0.01, *** *p* < 0.001).

**Figure 7 cancers-14-05368-f007:**
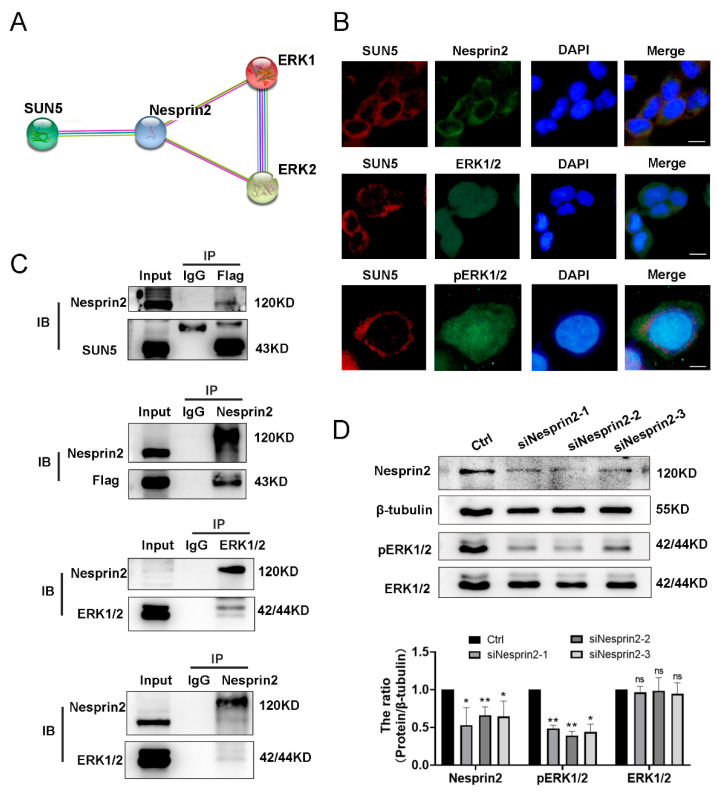
SUN5 cooperated with Nesprin2 to regulate the ERK pathway. (**A**) Interaction network analysis of SUN5, Nesprin2, and ERK1/2 by the STRING database. (**B**) Immunofluorescence staining results showed the co-localization of SUN5 (red), Nesprin2 (green), and ERK1/2 (green) in DLD-1 cells. DAPI was used for nuclear staining (blue). Scale bar: 15 μm, 15 μm, and 5 μm from top to bottom. (**C**) The interaction network of SUN5, Nesprin2, and ERK1/2 was validated by Co-IP analysis. (**D**) Western-blotting assay showed that Nesprin2 silencing decreased the level of pERK1/2 in SUN5-OE cells. β-tubulin was used as an internal control (* *p* < 0.05, ** *p* < 0.01; “ns” indicates not significantly different).

**Figure 8 cancers-14-05368-f008:**
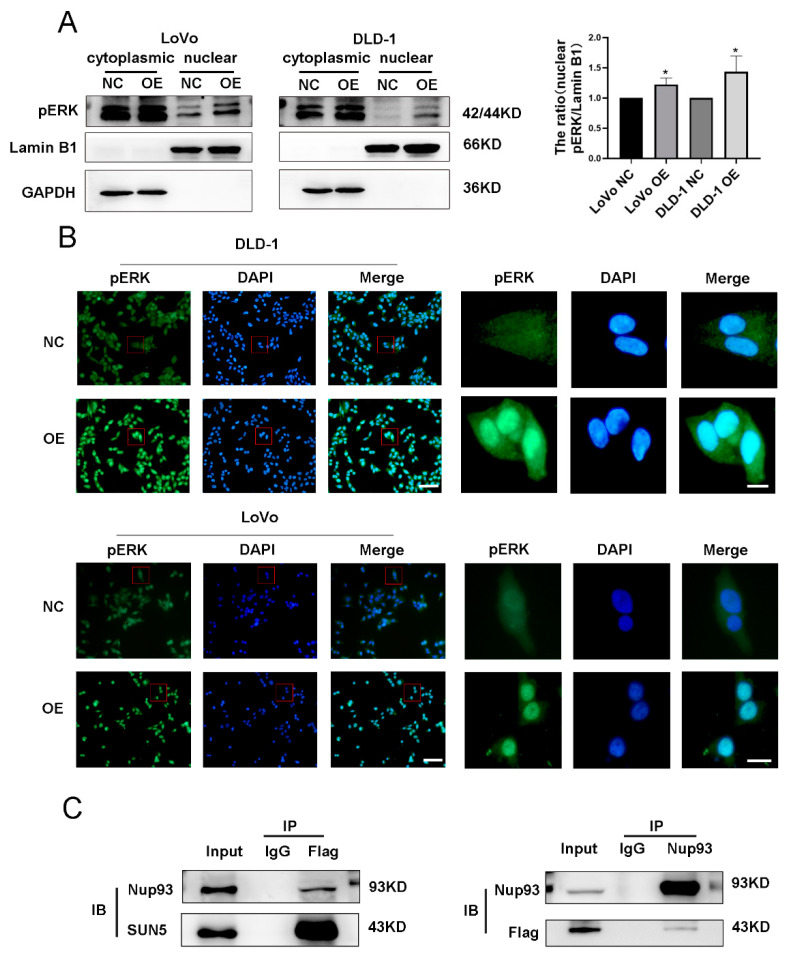
SUN5 interacted with Nup93 to promote phosphorylated ERK1/2 nuclear translocation. (**A**) The expression of pERK1/2 was assessed in SUN5-OE and NC cells by Western blotting. GAPDH and LaminB1 were used as controls for cytoplasmic and nucleus fractional purity, respectively (* *p* < 0.05). (**B**) The expression and subcellular localization of pERK (green) were determined by immunofluorescence staining. DAPI was used for nuclear staining (blue). Scale bar: 50 μm (left); 5 μm (right). (**C**) The interaction between SUN5 and Nup93 was validated in SUN5-OE cells by Co-IP analysis.

## Data Availability

Data required for bioinformatics analysis were downloaded from the GEO (GSE87211, accessed on 28 November 2020, https://www.ncbi.nlm.nih.gov/geo) and the STRNG database (accessed on 10 November 2020, https://cn.string-db.org/).

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
