# Peer review of "Nuclear Membrane Protein SUN5 Is Highly Expressed and Promotes Proliferation and Migration in Colorectal Cancer by Regulating the ERK Pathway"

_cancers, 2022, doi:10.3390/cancers14215368_

Round 1
Reviewer 1 Report
In the manuscript by Song et al. entitled “Nuclear membrane protein SUN5 is highly expressed and promotes proliferation and migration in colorectal cancer: regulating the ERK pathway” the authors studied the expression of the nuclear membrane protein SUN5 both in different cancer cell lines and in several colorectal cancer (CRC) tissue samples, showing for the first time that SUN5 is highly expressed in CRC. In addition, they studied the function of the protein in in vitro and in vivo models and proposed a novel molecular mechanism of SUN5, according to which the protein promotes the proliferation and migration of CRC cells by regulating the ERK pathway via Nesprin2.
Overall, this is a good manuscript. By using different approaches and models the authors produced a huge amount of data supporting the founding hypothesis of the paper and the proposed model. The original and novel data provide novel insights into the pathogenesis of CRC and a new potential diagnostic marker and therapeutic target for CRC.
Thus, to my opinion, the manuscript may be suitable for publication in Cancers journal. I have only a concern that is intended to improve the manuscript: a thorough revision of the language is required. There are a number of grammar errors; in addition, particular attention should be paid to changes in tense.
Author Response
Thank you for your comments. We have sent our manuscript to MDPI for English editing to improve the language. Thanks for your support of our manuscript again.
Reviewer 2 Report
Very interesting work. I did enjoy by reading the manuscript.
Author Response
Thanks for your comments. We have improved the quality of figures and edited the language. Thanks for your approval of our manuscript again.
Reviewer 3 Report
In this paper authors studied the role of SUN5 in colo-rectal cancer in vivo and in vitro. Their results showed that SUN5 is overexpressed in several cancers although mainly highly expressed in colon cancer cell lines in vitro and in the large part of patient’s CRC samples.
Hyper-expression of SUN 5 induces phospho-ERK1/2 while silencing yielded the opposite result. SUN 5 overexpression partially reversed MEK inhibitor effects.
RNA SEQ analysis of SUN 5 over-expressing cells showed, between DEGs, gene involved in signalling pathways such as MAPK, cell cycle, mTOR, Wnt, ubiquitin-mediated proteolysis, and Hippo signalling, which exert roles in proliferation, migration, and survival. The attention of authors has been particularly focused on MAPK pathway. Indeed, in cells overexpressing SUN5 authors showed increased ERK1/2 phosphorylation while decreased in SUN5 silenced cells. Tumour formation is increased by SUN5 overexpression has been also showed.
Authors by means of experiment of co-immunoprecipitation hypothesized the involvement of Nesprin 2 in the ERK1/2 nuclear translocation recruiting NUP93. The final conclusion is that SUN5 regulate CRC proliferation and migration by regulating ERK pathway.
The work is interesting particularly because is focused on the mechanisms by which proteins interacting with MAPK pathway play a prominent role in orchestrating the conditions to promote uncontrolled growth and tumour dissemination in CRC, helping the identification of new targets for innovative CRC anti-tumour therapies.
Several important data are obtained by western blots that authors showed without correct quantification and statistics. Authors did not even state the number of experiment done with same result.
Some points deserve to be raised as follows
- Fig 1 A: Is not clear which relative control, taken as 1, were used to evaluate the levels of SUN 5 of all cancer samples shown in the figure. Please specify in the legend.
- Fig .1C: the level of SUN5 in CRC and in normal adjacent tissues. For greater clarity, in the legend it would be better to indicate whether the sample of CRC is C? and that of relative adjacent tissue is N?
In addition these western blots were not quantitated. Authors analysed 40 matched pairs of human CRC tissues and adjacent non tumour tissues. Authors show only partial results in Fig. 1C. Since the results are only partial and in some cases the differences between C and N appears subtle by a simple view, it is better to present the quantitative evaluation of Blots and to make a statistic of the results from the 40 samples. This is important to validate the fact that CRC samples overexpress SUN5 and that SUN5 overexpression is not casual or a characteristics of a group of tumour from particular patients, then the disease severity could possibly be diagnosed on the basis of the level of SUN 5 expression. This evaluation, to my view, is important.
Fig. 1F: The localization of SUN5 at the rim of nucleus is not particularly evident except after printing of the three pictures of immunostaining. A slight increase of contrast makes this distribution more evident.
Fig.2A and D: Authors are invited to present quantitative evaluation of western blots with statistics in A and D. In A, left panel it is important to have an estimate of the extent of SUN5 overexpression that induces increases of cell cycle proteins expression (shown in D at left) , while in right panel A it is important to have an estimate of the extent of silencing that reduces the expression of cell cycle proteins (in right panel D).
Fig.2 Legend: In the legend of Fig 2 authors mentioned SUN5-OE and SUN5-KD LoVo and DDL1 that are not present in the Figure or in the text. Authors are invited to update the legend in order to read the figure and samples called with the same acronyms in figure, legend as well as well as in the text when necessary. In A NC is control and OE is overexpressing cells? Please specify in the legend. Control in some cases is Blank please normalize the denomination of control sample when possible.
Fig3: western Blots are presented and evaluated as single experiment. Authors should show statistics.
Fig4F: Western blot quantitative evaluatin and statistics are missing
Fig 5: In this figure Blank is untreated cells? There is a discrepancy between levels of phospho-ERK1/2 of DDL1 control cells of fig 4 F compared to fig 5 B. These figures present level of phospho-ERK1/2 of LoVo and DDL1 cells. In Fig 4 phospho-ERK1/2 of DDL1cells appears at very lower extent than those in Fig 5B.These evident differences might be the reason why the concentration of PD325901 to block ERK1/2 phosphorylation in DDL1 is substantially higher than those used to produce same inhibition in LoVo cells. Authors explain this discrepancy. The overexpression experiments seem to show variability incompatible with a comparison of the results. Fig5B and 4 F should have been done with cells coming from same overexpression experiment.
Fig 6: Authors shows the in vivo tumour growth by transplanting SUN5 overexpressing LoVo cells or control cells, demonstrating the increased tumour volume and weight due to SUN5 overexpression. Since CRC cell lines per se overexpress SUN5, it is legitimate to ask what happens if the transplanted cells are the SUN5-KO? Are they still tumorigenic? The lack of tumorigenicity and decrease of phospho-ERK1/2 in tumour from SUN5-KO samples would have proved that SUN5 is upstream to ERK signalling also in in vivo context. Western blots in 6H lack quantitative evaluation and statistics.
Fig 7 and Fig 8: Co-immunoprecipitation experiments aimed at demonstrating the proteins association are very critical and obtaining the same result for at least three times is very important for the credibility of the result. Authors should show three replicates.
Author Response
Thank you for your comments. According to your suggestions and questions, we have revised our manuscript. All Western blotting results in this study have been quantified and statistically analyzed. We have also provided the number of experimental replicates in the materials and methods. Thanks for your advice of our manuscript again. Please see the attachment for details.
